# The Impact of Gold Nanoparticles on Somatic Embryogenesis Using the Example of *Arabidopsis thaliana*

**DOI:** 10.3390/ijms241210356

**Published:** 2023-06-19

**Authors:** Kamila Godel-Jędrychowska, Anna Milewska-Hendel, Katarzyna Sala, Rafał Barański, Ewa Kurczyńska

**Affiliations:** 1Institute of Biology, Biotechnology and Environmental Protection, Faculty of Natural Sciences, University of Silesia in Katowice, Jagiellońska 28, 40-032 Katowice, Poland; kamila.godel-jedrychowska@us.edu.pl (K.G.-J.); anna.milewska@us.edu.pl (A.M.-H.); katarzyna.sala@us.edu.pl (K.S.); 2Department of Plant Biology and Biotechnology, Faculty of Biotechnology and Horticulture, University of Agriculture in Krakow, Al. Mickiewicza 21, 31-130 Kraków, Poland; r.baranski@urk.edu.pl

**Keywords:** arabinogalactan proteins, gold nanoparticles, histology, immunohistochemistry, morphology, pectins, somatic embryogenesis, organogenesis

## Abstract

Although the influence of nanoparticles (NPs) on developmental processes is better understood, little is known about their impact on somatic embryogenesis (SE). This process involves changes in the direction of cell differentiation. Thus, studying the effect of NPs on SE is essential to reveal their impact on cell fate. This study aimed to examine the influence of gold nanoparticles (Au NPs) with different surface charges on the SE of *35S:BBM Arabidopsis thaliana*, with particular emphasis on the spatiotemporal localization of pectic arabinogalactan proteins (AGPs) and extensin epitopes in cells changing the direction of their differentiation. The results show that under the influence of nanoparticles, the explant cells of *35S:BBM Arabidopsis thaliana* seedling origin did not enter the path of SE. Bulges and the formation of organ-like structures were observed in these explants, in contrast to the control, where somatic embryos developed. Additionally, spatiotemporal changes in the chemical composition of the cell walls during the culture were observed. Under the influence of Au NPs, the following effects were observed: (1) explant cells did not enter the SE pathway, (2) the impacts of Au NPs with different surface charges on the explants were variable, and (3) the compositions of the analyzed pectic AGPs and extensin epitopes were diverse in the cells with different developmental programs: SE (control) and non-SE (treated with Au NPs).

## 1. Introduction

Nanotechnology has been an integral part of everyday life (for review, see, e.g., [1,2]), as nanoparticles (NPs) are used, among others, in medicine, pharmacy, agriculture, and a wide array of industrial sectors. This means that NPs are present not only in the environment but also in living organisms, from bacteria (e.g., [3] and the literature therein), plants [4,5,6] (and the literature therein), animals [7] (and the literature therein) to the human body (e.g., [8,9]). The influence of NPs on the function of living organisms is increasingly analyzed because NPs can be useful and may also carry the risk of toxicity. The impact of NPs on plant growth and development is intensively studied, especially concerning crops, as nanotechnology was intended to increase their productivity [10,11] (and the literature therein). In recent years, the influence of NPs on various plant developmental processes has been studied [12,13,14,15,16]. The results indicate that the effect of NPs can be stimulating or harmful, including on plant growth and development.

The influence of NPs on in vitro culture, including somatic embryogenesis (SE), has also been analyzed. SE is used for studying the regulation of embryogenesis and embryo development but is also one of the methods of plant micropropagation used for commercial purposes. During this process, somatic embryos, resembling their zygotic counterparts, develop from non-zygotic cells directly or indirectly (through callus formation; for review, see [17,18]). Several genes have been identified to play roles in the induction of embryogenesis in plants; for example, ectopic expression of the *BBM* gene induces spontaneous embryo formation. As a result, overexpression of the *BBM* gene in *Arabidopsis thaliana* is used in SE studies due to the easy way of obtaining somatic embryos [19]. Research on the correlation between NPs and SE is focused mainly on the effect of NPs on the efficiency of SE and the NPs’ influence on the physiological processes that occur at the various stages of SE (e.g., [20,21,22,23,24,25]). These studies so far indicate that NPs in plant tissue cultures give positive results and enhance plant regeneration (e.g., [23,26]).

Despite the increasing number of studies on the influence of NPs on SE, no data (to the best of the authors’ knowledge) have shown how NPs affect the cell wall during cell fate changes from the somatic to the embryogenic/meristematic state. In SE, somatic cells undergo restructuring to generate embryogenic cells, including the dedifferentiation of non-zygotic cells, the activation of cell division, and the reprogramming of the cell fate from the molecular to the histological level (for review, see [27]). As the process of SE involves changes in the developmental programs, it is crucial to know how NPs influence the process of cell differentiation and changes in the direction of differentiation.

The cell wall is no longer considered a dead structural element of plant cells but part of the cell that is precisely rearranged in response to biotic and abiotic factors [28,29,30,31,32,33,34]. Changes in the structure and chemical composition, including degradation, synthesis, and the deposition of new macromolecules of the wall, reflect the adaptation of plant cells to different conditions during plant growth and development, including SE [33]. Changes in the chemical composition of the cell walls concern their main components, such as polysaccharides and proteins, including arabinogalactan proteins (AGPs), pectins, and extensins [35,36]. For example, the embryogenic calli of *Brachypodium distachyon* were characterized by the occurrence of AGP epitopes recognized by the JIM16 and LM2 antibodies, an extensin epitope recognized by the JIM11 antibody, and pectic epitopes recognized by the LM6 antibody [35]. During the SE of *Arabidopsis thaliana*, it was shown that the AGP epitope recognized by the LM2 antibody could serve as a negative marker for embryogenic cells and a positive one for meristematic cells [33]. Analyses of the presence of different AGPs, pectins, and extensins in non- and organogenic calli of *Actinidia arguta* revealed that pectic epitopes recognized by LM5 and LM6 antibodies and the AGPs recognized by LM2 antibodies were detected only in the non-organogenic calli [31]. Studies on cell wall composition during *Daucus carota* SE showed that, among others, the LM6 (pectic) and LM2 (AGP) epitopes were positive markers, but the LM5 (pectic) and JIM13 (AGP) epitopes were negative markers of cell reprogramming to the meristematic/pluripotent state [34]. Therefore, undertaking research that concerns the influence of NPs on cell wall composition during SE is justified.

The influence of NPs may depend on their physicochemical properties, including their surface charge [37]. Many data indicate that the surface charge has a diverse impact on plant development [38,39,40,41,42,43]. Nevertheless, there are no data on the effect of NPs’ charge on SE. Therefore, the aims of this study were: (1) to analyze the impact of Au NPs with positive, negative, and neutral charges on the SE process, and (2) to answer the question of whether Au NPs affect SE, not only its efficiency but also the composition of the cell walls of explant cells of the *35S:BBM Arabidopsis thaliana* transgenic line during the transition from the somatic to the embryogenic and meristematic state.

## 2. Results

### 2.1. Developmental Changes in Explants during Culture

The analyses presented in this study include two time points: after six days (early stage) and after 14 days (late stage) of culture duration. The seedling explants of *35S:BBM* were characterized by the formation of an embryogenic margin on the periphery of the cotyledons, within which embryogenic protrusions appeared (Figure 1A inset; description of margin cells and protrusions is in [44]). At the early stage, the explants in the control conditions had well-developed cotyledons with margin cells at the cotyledon rim, hypocotyl, and small roots (Figure 1A; arrows). The explants treated with BPEI and citrate Au NPs were developmentally similar (Figure 1B,C), with no visible morphological changes on the cotyledons. The PEG Au NP-treated explants were the most delayed in development at the early stage of culture compared to the rest of the treated and control explants (Figure 1D). The explants after treatment with BPEI Au NPs were characterized by elongated hypocotyls (Figure 1B) and showed delayed development compared to the control explants. In the case of the PEG Au NP-treated explants, malformations at the base of the cotyledons were observed at the early stage of culture (Figure 1D inset). At the late stage of culture, the control explants showed typical morphology with the development of margins at the periphery of the cotyledons, protrusions, and somatic embryos (Figure 1E and insets and 1I). Somatic embryos were not detected on the explants treated with the Au NPs (Figure 1F,G,J–L) apart from a tiny amount in the explants treated with BPEI Au NPs (Figure 1F; Table 1). The formation of organ-like structures on the hypocotyl (Figure 1F–H) or cotyledons (Figure 1J–L) of the explant was observed in the Au NP-treated explants. Organ-like structures may resemble leaves (Figure 1G), somatic embryos (Figure 1H; because the vascular tissue not separated from the explant is not a somatic embryo), and cotyledons (Figure 1J–L; the organ-like structure characteristic is representative of all figures shown). Statistically significant differences in development were detected only between the PEG-treated and the other Au NP-treated explants (Table 1).

To summarize, the treatment of explants with the Au NPs resulted in the formation of organ-like structures (Figure 1), and the detected differences between the treated and the control explants were statistically significant (Table 1). In general, the explants exposed to Au NPs exhibited delayed development compared to the control, which was manifested by the absence of visible margins around the cotyledons and a retarded response to the growing conditions. At the late stage of culture, the formation of somatic embryos was not detected, which means that the process of SE was blocked. However, in the explants treated with Au NPs, the organogenesis processes occurred, in which not only the cotyledons were involved but also the hypocotyl.

### 2.2. Histological Changes in Explants

At the early stage of culture, the first changes occurred in the control explants. This time point was selected based on previous studies, which have shown that changes in the direction of cell differentiation occur for up to seven days of culture. After two weeks, somatic embryos are well-developed [44]. At the early stage of culture, the first visible sign on the control explants was the appearance of a single layer of margin cells characterized by an isodiametric shape and the presence of a large nucleus with one nucleolus increased in size (Figure 2A and inset). These cells represented the first evidence of changes in cell differentiation. In the following days, these cells formed protrusions that consisted of embryogenic cells (Figure 2B and inset). At the late stage of culture, somatic embryos were present (Figure 2C). In the Au NP-treated explants, margin cells or protrusion formation were not observed, but the appearance of surface cells (1 to 3 layers of explant cells with a phenotype of meristematic character; Figure 2D,H) and bulges (Figure 2E,I; resulting from the local divisions of explant cells) were observed. The surface cells were meristematic (Figure 2D and inset). These cells were characterized by a dense cytoplasm, small vacuoles, starch grains, and the cell nucleus containing one or two nucleoli (Figure 2D and inset). Such features are typical for meristematic cells. The meristematic cells formed a bulge that differed from the other explant regions (Figure 2E and inset and Figure 2I). On the 14th day, from the bulge, organ-like structures (resembling leaves) developed (Figure 2F and inset and Figure 2G).

### 2.3. Spatio-Temporal Changes in the Wall Components

Studies of the chemical compositions of the cell walls were carried out within the explant: in cells that did not change their direction of differentiation (described further as explant cells), in cells that changed their direction of differentiation (embryogenic and meristematic cells, for description, see [44]), and in somatic embryos and organ-like structures. As described above, changes in the Au NP-treated explants appeared on cotyledons and hypocotyls. However, for the analysis of the chemical compositions of the cell walls, only changes in the cotyledons were compared, as only such changes occurred in the control explants.

### 2.4. Distribution of Pectic Epitopes during Culture

The LM5 antibody (which recognizes the galactan side chains of rhamnogalacturonan I, RG I) was detected in the control, BPEI, and citrate Au NP-treated explants (Table 2; Figure 3A–D) at the early stage. This pectic epitope was abundant in the control explants (Figure 3A inset) and protrusions (Figure 3B) as well as in most Au NP-treated explants. However, in the PEG Au NP-treated explants, this epitope was practically absent, apart from some individual walls (Figure 3E). At the late stage of culture, the LM5 epitope was detected in the somatic embryos of the control (Figure 3F). It was also present in the surface cells and bulges of the BPEI Au NP-treated explants, mainly in new walls as a result of the cytokinesis of the explant cells leading to protrusions (Figure 3G). However, it was not detected in the citrate-treated explants, including the surface cells and bulges (Figure 3H). In the PEG Au NP-treated explants, the LM5 epitope was observed in the surface cells and organ-like structures (Figure 3I).

The pectic epitope recognized by the LM6 antibody (arabinan side chains of RG I) was abundantly present in the walls of all explant cells regardless of the type of treatment (Table 2; Figure 3J–M) at the early stage. In the control explants at the late stage, LM6 was observed in the margin cells and protrusions but not detected in the somatic embryos (Figure 3N). In the BPEI Au NP-treated explants, the presence of LM6 was observed in the explant cells and organ-like structures (Figure 3O and inset). However, no LM6 signal was detected in the surface cells (cells changing their direction of differentiation; Figure 3O). In the citrate Au NP-treated explants, the LM6 epitope was observed in the explants, surface cells, bulges, and organ-like structures (Figure 3P). In the PEG Au NP-treated explants, an LM6 fluorescent signal was present in the vascular bundle, some parenchymatous cells, and a few epidermal cells (Figure 3Q; the negative control exhibited no fluorescence signal independent of the used antibodies and confirmed the specificity of a secondary antibody, as shown in Appendix A).

### 2.5. Distribution of AGP Epitopes during Culture

Epitopes detected by the JIM13 antibody were present in the vascular bundles in all types of explants (control and Au NP-treated) at the early stage of culture (Table 2; Figure 4A–D) and in the margin cells of the control explants (Figure 4A). A JIM13 signal was not observed in the other explants cells in any of the analyzed variants at any stage (Figure 4A–G). JIM13 did not appear in the margin cells or protrusions in the control explants (Figure 4E) nor in the bulges or organ-like structures in the Au NP-treated explants (Figure 4F,G).

The presence of AGPs recognized by the MAC207 antibody was diverse in the different types of explants at the early stage (Table 2; Figure 4H–K). In the control, BPEI, and PEG Au NP-treated explants, MAC207 was not detected (Table 2; Figure 4H,I,K), in contrast to the explants treated with citrate Au NPs, where MAC207 epitopes occurred abundantly (Figure 4J). At the late stage of culture, the MAC207 epitope was observed only in somatic embryos (Figure 4L). In the explants treated with BPEI Au NPs, this epitope was not detected in an organ-like structure (Figure 4M) and in bulges (Figure 4N) but was visible only in the explant cells. The explants treated with citrate Au NPs were characterized by this abundant epitope presence in the explant cells’ cytoplasmic compartments (Figure 4O and inset). In the PEG Au NP-treated explants, this epitope was detected in the explant cells and organ-like structures (Figure 4P).

Other AGP epitopes recognized by the LM2 antibody were present in the explants regardless of the treatment (Table 2; Figure 5A–D) at the early stage of culture, including margin cells and protrusions in the control (Figure 5A inset). At the late stage of culture, the LM2 epitope signal was present in the control explant and margin cells, protrusions, and somatic embryos (Figure 5E and inset). In the BPEI and citrate Au NP-treated explants, LM2 occurred in the explant cells (Figure 5F inset, Figure 5G inset). Furthermore, LM2 was detected in the surface cells and bulges of BPEI Au NP-treated explants (Figure 5F inset) but not in the organ-like structures (Figure 5F,G). In the PEG Au NP-treated explants, the signal was observed in the organ-like structures and the explant cells (Figure 5H).

AGPs recognized by the JIM16 antibody were detected in the control explants at the early stage, except for the cells that changed their direction of differentiation, i.e., margin cells. In these cells, a noticeable signal was observed only in the outer periclinal walls of the epidermal cell (Figure 5I). In the Au NP-treated explants, JIM16 was also present (Figure 5J–L). In the BPEI-treated explants, the cell signal was detected in the cytoplasmic compartments (Figure 5J), but in the citrate- and PEG-treated explants, the signal was detected mainly in the walls of the cells (Figure 5K,L). At the late stage of culture, the JIM16 epitope was observed in the explant cells and the protrusions, and somatic embryos of the control explants (Figure 5M). In the BPEI Au NP-treated explants, JIM16 occurred in the explants, surface cells, bulges, and organ-like structures (Figure 5N). The citrate Au NP-treated explants did not exhibit the presence of the JIM16 epitope (Figure 5O). In the PEG Au NP-treated explants, this epitope was observed in the bulges (Table 2) and organ-like structures (Figure 5P).

### 2.6. Distribution of Extensin Epitopes during Culture

Extensins recognized by the JIM11 antibody were not observed in the explant cells in either the control or treated explants at the early stage of culture (Figure 6A–C, insets). At the late stage, the epitope JIM11 was visible only in the explant cells of the control, BPEI, and citrate Au NP-treated explants (Figure 6D–F) as well as in the vascular bundle in the PEG Au NP-treated explants (Figure 6G). Importantly, this epitope was not observed in the cells that changed their direction of differentiation (Figure 6D–G).

## 3. Discussion

### 3.1. Nanoparticles and Somatic Embryogenesis

The presented results concern the influence of NPs on developmental processes during the SE of the *35S:BBM Arabidopsis thaliana* transgenic line. In the regulation of in vitro culture, many factors are involved, including, among others, cell wall composition alteration, cell-to-cell signaling, hormonal changes, and epigenetic shifts [17,18,44].

The results presented here indicate that the SE was blocked by the Au NP treatment, regardless of the surface charge used. The literature data overwhelmingly indicate the opposite effect of using NPs on SE. For example, in studies on *Ocimum basilicum* [23], *Teconella undulata* [45,46], *Linum usitatissimum* [22], *Musa paradisiacal* [47], *Gloriosa superba* [48], *Coffea arabica* [49], and *Panax vietnamensis* [50], NPs improved the efficiency of SE. Data on SE inhibition under the influence of NPs are available but scarce. Experiments with *Daucus carota* culture treated with Fe_3_O_4_ NPs showed no embryo formation, indicating that the NPs blocked the embryogenetic pathway [25]. Studies on *Panax vietnamensis* showed that Ag NPs, at a concentration of 2.0 mg/L, inhibited somatic embryo formation [51]. A decreased number of somatic embryos was noticed for *Phoenix dactylifera* after treatment with carbon nanotubes [52]. Our results are the next example of the inhibitory effect of NPs on SE. It can be concluded that inhibition of the SE by Au NPs may indicate their influence on cell differentiation and the implementation of the embryogenic program.

### 3.2. Cell Wall Composition during SE in Control

Changes in cell fate and differentiation, including SE, are accompanied by the controlled remodeling of the cell wall polymer networks necessary to permit cell division and expansion [53,54,55,56,57,58,59,60,61,62]. Alterations in the chemical composition of the cell walls are used as markers of the cell response to various biotic and abiotic factors and as markers of changes in the cell fate during in vitro cultures, including SE and organogenesis (for review, see [28,62]).

In the present studies, the distribution of some pectic, AGP, and extensin epitopes during the SE of the *35S:BBM* transgenic line of *Arabidopsis thaliana* was analyzed. This line has been thoroughly studied at the molecular [19] and symplasmic communication levels, and as a source of somatic embryos [44], but there is no information about changes in the composition of the cell walls during the SE of this line.

The obtained results show spatiotemporal changes in the wall composition during SE. Pectic epitopes recognized by the LM5 and LM6 antibodies were present in all explant cells from the acquisition of embryogenic competence to the end of the culture. On this basis, it can be concluded that these pectic epitopes were not markers of embryogenic cells. On the contrary, studies on *D. carota* SE showed that the LM5 epitope is a negative marker of cells reprogramming to the meristematic/pluripotent state, and the LM6 epitope is a positive marker of SE [34]. Similar results were described for *Trifolium nigrescens* explants, where the LM5 epitope was not observed in embryogenic protrusions, but the LM6 epitope was detected predominantly in the cells of embryogenic swellings [63]. However, during the SE of *Arabidopsis thaliana* Col-0, the LM5 epitope was abundant in the walls of cells localized within the embryogenic domain [33]. The exact function of the pectin arabinan and galactan side chains of RG-I remains unknown. However, the galactan side chains of RG-I are postulated as a component that maintains cell stiffness, and the arabinans likely fulfill the function of pectic plasticizers in maintaining cell wall flexibility [64]. Due to a few studies analyzing changes in the presence of LM5 and LM6 epitopes, especially during reprogramming from the somatic to the embryogenic state, no general conclusions can be drawn. However, it can be stated that in the *35S:BBM Arabidopsis thaliana* transgenic line, arabinans and galactans are not markers of changes in the direction of cell differentiation.

AGPs are involved in many developmental processes, including SE [64]. Among the AGP epitopes analyzed here, only the epitope recognized by the JIM13 antibody can be a positive marker of cells undergoing the induction of SE in the control explants. Similar results, for example, were described for *Agave tequilana* SE [65], microspore embryogenesis of *Brassica napus* [66], or *Centaurium erythraea* SE, where the JIM13 epitope was detected in embryogenic cells [67]. According to the obtained results, the AGPs recognized by the JIM16 antibody can serve as a negative marker of cells during the acquisition of embryogenic competence because it was not detected in the margin and protrusion cells at the early stage of the culture. Opposite results were described during the SE of *Musa* spp [68], *Arabidopsis thaliana* [33], and *Brachypodium distahyon* [35], where JIM16 epitopes were strongly expressed in embryogenic cells. The next epitope analyzed here was LM2, which was found to be a positive marker of embryogenic cells during SE. Similar results were described, for example, for *Brachypodium distachyon* embryogenic calli [35], *Centaurium erythraea* [67], and *Zea mays* SE [69] or *Trifolium nigrescens* [70]. On the contrary, during the SE of *Arabidopsis thaliana* Col-0, the LM2 was postulated as a negative marker for cells changing their fate from the somatic to the embryogenic state [33].

The extensin recognized by the JIM11 antibody was detected only in explant cells on the 14th day of culture, which indicates that this epitope was not involved in the cell reprogramming. The presence of extensins in explants during in vitro culture has not been widely analyzed. It has been detected in *Dactylis glomerata* [71,72], *Phalenopsis* [73], *Musa* spp. [68], and *Brachypodium distachyon* [35] in in vitro cultures. Thus, the involvement of extensin in SE may be species-dependent, but much more research needs to be done. Our results indicate that the JIM11 epitope is not a marker of a change in the direction of cell differentiation during the SE of the *35S:BBM Arabidopsis thaliana* transgenic line. The short review of the literature data presented above indicates that none of the tested epitopes stands out as a universal marker of SE, and the reasons for this are widely discussed, pointing to, among others, species dependence, culture conditions, and explant sources [74].

### 3.3. Differences in Wall Composition between Control and NP-Treated Explants

The second aspect of the presented studies relates to changes in the chemical composition of the walls of the explant cells during the in vitro culture under treatment with Au NPs of different surface charges and comparison to the control explants. Different chemical compositions of the explant cell walls implementing various developmental programs under the influence of Au NPs were found (Table 2).

The chemical compositions of the cell walls of the explants treated with Au NPs varied depending on the charge of the nanoparticles used (Table 2). This variation applied to both pectin and AGP epitopes. In the surface cell, the AGP epitope was found only in the BPEI Au NP-treated explants. In the walls of the surface cells of the explants treated with citrate Au NPs, only the LM6 epitope was present. Similar diversity was found in the walls of the bulges, although in comparison to the surface cells, AGPs were detected. The organ-like structure also lacked some analyzed epitopes (Table 2). In addition, a comparison of the chemical composition of the walls of cells involved in SE (margin cells, protrusions, and somatic embryos) with those of cells not implementing the SE program (surface cells, bulges, and organ-like structures) showed that, in the latter case, pectic and AGP epitope distribution decreased.

It is postulated that pectins are involved in many developmental processes, from cell differentiation to the reaction of plants to biotic and abiotic stresses [32] (and the literature therein). The qualitative composition of pectins within the wall under the influence of NPs is little known, and the effect of NPs on the composition of cell walls during in vitro cultures, including SE, is not described at all. The available literature data show that in the roots of *Arabidopsis thaliana* seedlings, nano zerovalent iron induces the loosening of the cell walls [75] and that CuO NPs cause physical damage and a biochemical disruption of the cell walls [76]. This indicates that remodeling of the cell walls might be the first reaction to the presence of NPs.

Among the analyzed pectic epitopes, a diverse presence of arabinan and galactan was detected. These epitopes were detected in embryogenic cells, which permit cell division and expansion [33,34,35,45,56,57,60,61,62]. In the explants treated with neutral and positively charged Au NPs in the cell walls, only the galactan epitope was detected, and in the explants treated with negatively charged Au NPs, the cell walls were rich in arabinans. At the current research stage, it is unknown what could be the mechanism of such a different response of cells to NPs.

AGPs are postulated to be involved in different developmental processes and plants’ reactions to environmental factors [33] (and the literature therein). The available information concerns the effect of NPs on plant growth and development. Still, no papers describe the impact of NPs on the processes of changing the direction of cell differentiation (such as the SE process) and the accompanying changes in the chemical composition of the walls (to the best of our knowledge).

It is worth mentioning that extensins recognized by the JIM11 antibody were detected only in the walls of the explant cells, which did not change the developmental program but increased in size, which supports the hypothesis that the extensins may play a role in cell expansion [77,78]. The influence of NPs on cell wall remodeling is a fact, but as the extensin role has not been studied yet, this finding opens up new possibilities for studying the correlation between NPs and extensins.

The differences between the pectic, AGP, and extensin epitopes in the walls of cells that underwent different developmental programs under the influence of Au NPs probably depended on the surface charge of the NPs used. It is postulated that positively charged NPs tend to adhere to the organ surface, especially the roots, but negatively and neutrally charged NPs enter the plant body more easily [32,38,39,40,41,42,43,79,80,81,82]. The results of our research indicate that regardless of whether the NPs overcame the wall and plasmalemma barriers or whether they remained on the surface of organs, they caused changes in the chemical composition of the walls. The influence of Au NPs on wall composition was described so far only for *Hordeum vulgare* and showed differences in the wall composition of root cells depending on the surface charge of NPs [32].

The influence of NPs, including Au NPs, on the chemical composition of cell walls has not been extensively studied, especially in the context of their impact on the chemical composition of cell walls implementing different development programs, including SE. Thus, this is probably the first report showing the qualitative changes to the individual components of the walls of cells implementing various developmental programs and the impact of NPs on this process.

## 4. Materials and Methods

### 4.1. Nanoparticle Characterization

Gold nanoparticles (spheres), 5 nm in diameter (nanoComposix Europe, Prague, Czech Republic), were coated with (1) polyethylene glycol (PEG), which neutralizes charge; (2) branched polyethyleneimine (BPEI), which contains amino groups, which cause the formation of positively charged Au NPs; and (3) trisodium citrate, which causes the formation of negatively charged Au NPs. The designations PEG Au NPs, BPEI Au NPs, and citrate Au NPs were adopted for the individual surface charge of the NPs used, and in the figures, the following nomenclature is used: PEG, BPEI, and citrate, respectively.

### 4.2. Plant Material and Growth Conditions

The material used for the study was the *Arabidopsis thaliana* (L.) Heynh Columbia-0 *35S:BBM* transgenic line, which is characterized by the spontaneous formation of somatic embryos on the cotyledons [19,44]. Seeds of *35S:BBM* were surface-sterilized with 20% commercial bleach sodium hypochlorite (ACE Lever Co., Fater S.P.A., Pescara, Italy) for 10 min and then washed 4–5 times (5 min each) with sterile distilled water, left for 5 min in 70% ethanol, and then again washed with sterile water. The seeds were left in darkness in sterile water overnight at 4 °C. Then the seeds were placed in an Erlenmeyer flask with liquid half-strength Murashige and Skoog (MS) medium containing 1% sucrose, pH = 5.8 [83]. The seeds were germinated in control conditions (1/2 MS medium) and with the addition of three different types of Au NP solutions (neutral, positively, and negatively charged) at a concentration of 25 μg/mL. Liquid (rotary shaker, 60 rpm/min) cultures were kept at 21 °C (16 h light/8 h dark with a light intensity of 40 μmol m^−2^ s^−1^) for two weeks. The plant material was collected on the 6th (early stage of culture) and 14th (late stage of culture) days of culture.

### 4.3. Sample Preparation

The samples were fixed in a mixture of 4% paraformaldehyde and 2% glutaraldehyde 0.1 M phosphate-buffered saline (PBS; pH = 7.2) at 4 °C for 24 h. Then the samples were washed in PBS (pH = 7.2), dehydrated in a graded ethanol series, and embedded in Steedman’s wax [29]. Sections (7 μm thick) were cut using a HYRAX M40 rotary microtome (Zeiss, Oberkochen, Germany) and collected on poly-L-lysine-coated microscope slides (Menzel-Glaser, Braunschweig, Germany). The sections were de-waxed and rehydrated in a successive ethanol series. For histochemical analyses, the sections were stained with 0,05% Toluidine Blue O aqueous solution for 2 min. For immunohistochemistry, the sections were outlined with a hydrophobic marker (PAP pen; Sigma-Aldrich, Saint Louis, MO, USA) and submerged in a blocking buffer containing 2% (*v*/*v*) fetal calf serum (FCS) and 2% (*m*/*v*) bovine serum albumin (BSA) in PBS, pH 7.2, at room temperature for 30 min. The primary monoclonal antibodies (Plant Probes, Leeds, UK) were diluted 1:20 in blocking buffer at room temperature for at least 1.5 h. The antibodies used in the present studies are listed in Table 3. AlexaFluor 488 goat anti-rat (Jackson ImmunoResearch Laboratories, West Grove, PA, USA) diluted 1:100 in a blocking buffer was used as a secondary antibody. The cell wall was stained with 0.01% (*w*/*v*) calcofluor white (Fluorescent Brightener 28; Sigma-Aldrich) in PBS for 10 min. The negative controls were prepared by omitting the application of the primary antibody. Fluoromount™-mounted sections were analyzed using a Nikon Eclipse Ni-U microscope equipped with a Nikon Digital DS-Fi1-U3 camera with the corresponding software (Nikon, Tokyo, Japan) at a maximum excitation wavelength of 490 nm (AlexaFluor 488) or 330 nm (calcofluor white). The epifluorescence microscopy images were prepared as figures using Corel Draw x12 and Corel Photo-Paint software (the brightness and contrast were adjusted).

### 4.4. Statistical Analysis

The seedlings exhibiting and lacking developmental changes were counted, and a contingency table for the control and NP treatments was analyzed. The Pearson χ^2^ test was applied to test the significance of the NP treatments, assuming at least a 0.05 significance level. The final results were expressed as the percentages of the seedlings with developmental changes.

## 5. Conclusions

We analyzed the effect of Au NPs with different surface charges on SE and the chemical composition of the walls in cells changing their direction of differentiation and entering the path of SE, and those that did not enter the SE path. Studies have shown that: Au NPs (1) block the embryogenic pathway, (2) influence the chemical composition of the cell walls to the analyzed pectic (LM5, LM6), AGP (JIM13, MAC207, LM2, JIM16) and extensin (JIM11) epitopes, (3) and the surface charge of Au NPs has a significant effect on the chemical composition of the cell wall. So far, there have been no comprehensive studies on the impact of NPs on SE in the context of cell reactions during the changes in their direction of differentiation under the influence of NPs with different surface charges. The obtained results and their comparison with the literature data make evident the need for further studies to understand the effects of different NPs on developmental processes to explore and understand the intrinsic properties of NPs.

## Figures and Tables

**Figure 1 ijms-24-10356-f001:**
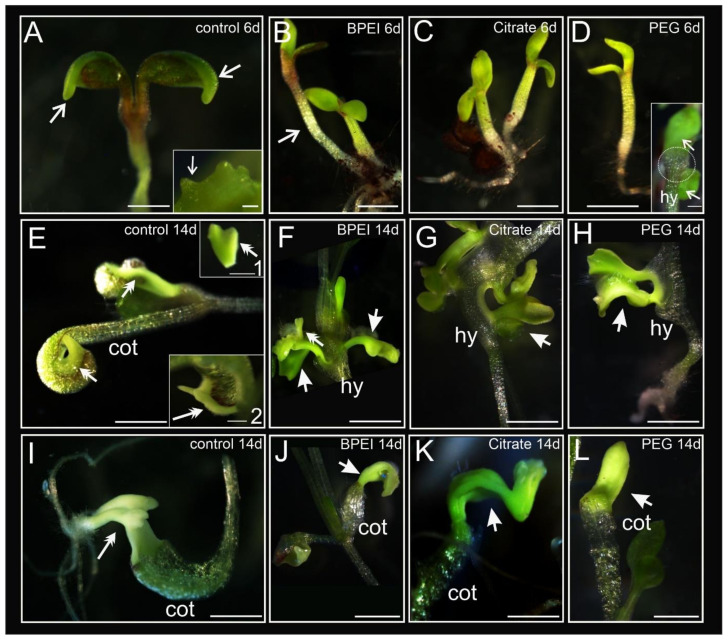
Morphological changes during in vitro culture of the control and Au NP-treated explants. (**A**) Control seedling explant with embryogenic margin cells (arrows); inset—protrusions that were developed from margin cells (arrow). (**B**) BPEI Au NP-treated seedlings; open arrow—elongated hypocotyl. (**C**) Citrate Au NP-treated explants. (**D**) PEG Au NP-treated seedlings explants; inset—malformations of the cotyledonary node (white dotted circle) at the base of cotyledons (arrows). (**E**) Control seedling explants with visible margin cells and somatic embryos in different stages of development (double arrows); insets 1 and 2—bipolar somatic embryos (double arrows). (**F**–**H**) Organ-like structures formed on the hypocotyl of Au NP-treated explants (arrows); the double arrow on F indicates a somatic embryo. (**I**) Bipolar somatic embryos in the torpedo stage developed on the adaxial part of cotyledons of the control seedling explant (double arrow). (**J**–**L**) Organ-like structures formed on the cotyledons of Au NP-treated explants (arrows). Cot—cotyledon; hy—hypocotyl; 6d—6th day of the culture; 14d—14th day of the culture, NP—nanoparticle. Scale bars: (**A**,**E**) 1 mm; A inset—50 µm; (**F**–**L**) D inset, E insets 1 and 2—500 µm; (**B**–**D**) 2 mm. The analyses were carried out using a stereo microscope.

**Figure 2 ijms-24-10356-f002:**
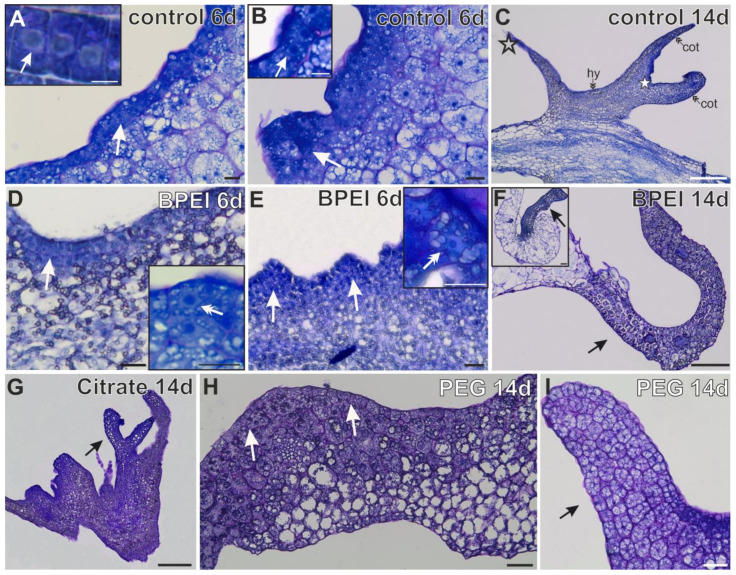
Histological analysis during the culture of the control and Au NP-treated explants. (**A**) Margin with embryogenic cells (arrow); inset—higher magnification of margin cells with a visible nucleus and nucleolus (arrow) in control explants. (**B**) Protrusions (arrow), inset—embryogenic cells from the area of protrusions (arrow). (**C**) Somatic embryo in control explants (cot—cotyledon; hy—hypocotyl; empty asterisk—embryogenic root; white asterisk—SAM). (**D**) Surface cells (arrow); inset—higher magnification of meristematic cells (double arrow) in BPEI Au NP-treated seedlings. (**E**) Small bulges (arrows) consisting of meristematic cells (inset; double arrow). F and F inset—organ-like structures resemble cotyledons (**G**) Organ-like structures resembling somatic embryos (arrows). (**H**) Surface cells (arrows). (**I**) Big bulge (arrow); 6d—6th day of the culture; 14d—14th day of the culture Scale bar: (**A**) A inset, (**B**) B inset, D inset, E inset-10 µm; (**C**,**F**,**G**)- 200 µm; (**D**,**E**,**H**,**I**)- 30 µm; F inset-100 µm. The analyses were carried out using a light microscope of sections tangential to the surface of the explant.

**Figure 3 ijms-24-10356-f003:**
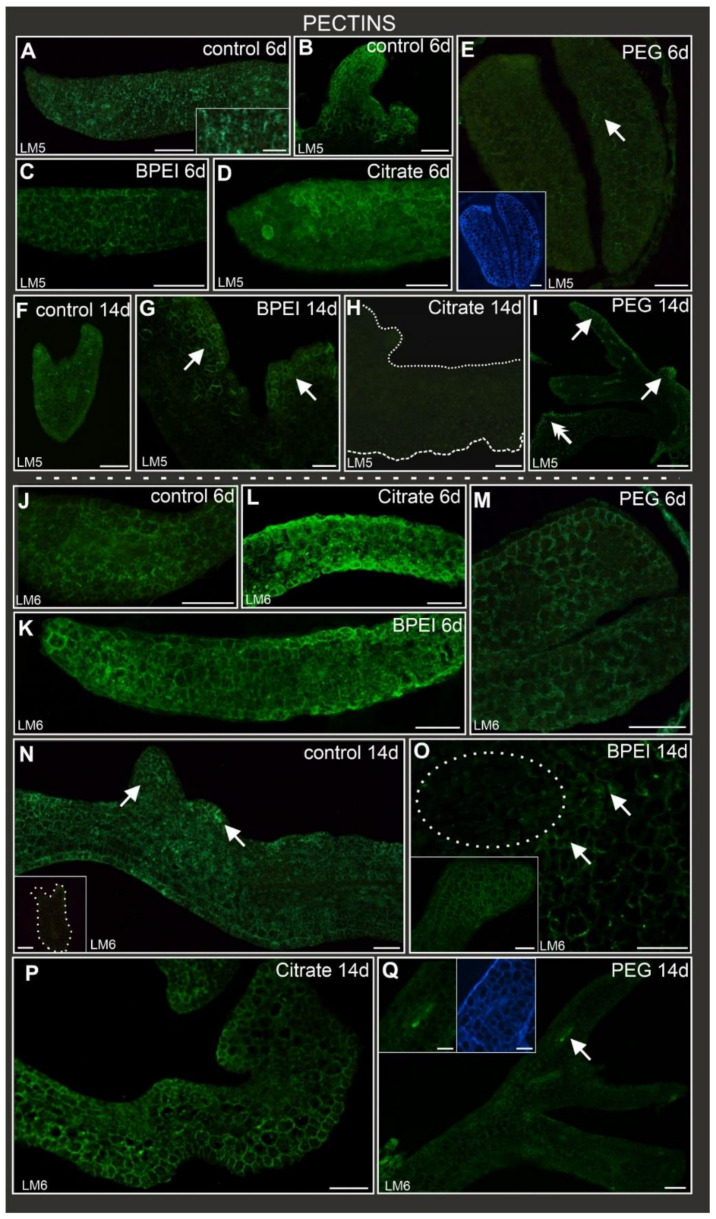
Localization of LM5 (**A**–**I**) and LM6 (**J**–**Q**) pectic epitopes in control and Au NP-treated explants. LM5 epitope present in: (**A**) control explants cotyledon, inset—higher magnification; (**B**) protrusions; (**C**) BPEI Au NP-treated cotyledons; (**D**) citrate-treated cotyledons. LM5 epitope absent from: (**E**) cotyledons, inset—calcofluor white section. (**F**) LM5 epitope present in somatic embryo; (**G**) LM5 detected in surface cells and bulges (arrows). (**H**) No LM5 signal in citrate Au NP-treated explant; dotted line delimits the border of the explant; (**I**) LM5 signal detected in surface cells (double arrow), bulges, and organ-like structures (arrows). LM6 epitope present in: (**J**–**M**) cotyledons; (**N**) margin, protrusions (arrows) and explant cells; inset—no LM6 signal in the somatic embryo (dotted line mark the embryo). (**O**) LM6 signal detected only in explant cells (arrows) without signal in surface cells and bulges (dotted white circle); inset—LM6 signal in organ-like structures; (**P**) LM6 signal in organ-like structures; (**Q**) organ-like structure with LM6 fluorescence; inset—higher magnification and calcofluor staining; arrow points to vascular cells. 6d—6th day of the culture; 14d—14th day of the culture. Scale bar: (**A**) A inset, (**B**,**D**,**E**), E inset—250 µm, (**G**)—50 µm; (**C**)—150 µm; (**F**,**H**,**I**,**J**,**L**), L inset, (**N**,**O**,**P**), Q inset—100 µm; (**K**,**M**,**Q**) 200 µm.

**Figure 4 ijms-24-10356-f004:**
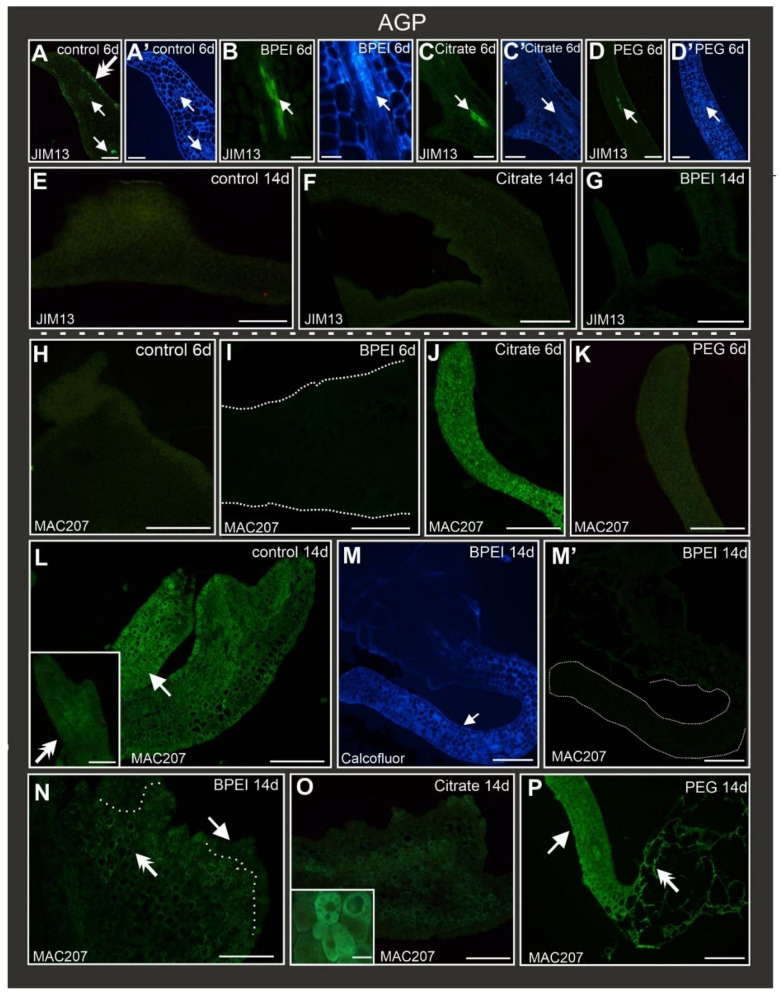
Localization of AGPs recognized by JIM13 (**A**–**G**) and MAC207 (**H**–**Q**) antibodies in control and Au NP-treated explants. (**A**) JIM13 epitope detected in margin cells (double arrow) and in vascular bundle (arrow) of control explant. JIM13 was detected in a vascular bundle in: (**B**–**D**) (arrows) all types of explants. (**A’**–**D’**) show the same section from (**A**–**D**) stained with calcofluor white. No JIM13 signal at the late stage: (**E**–**G**). Absence of MAC207 signal in explants: (**H**) control, (**I**) BPEI AuNP-treated, and (**K**) PEG Au NP-treated; MAC207 present abundantly in (**J**) citrate Au NP-treated explants. (**L**) Presence of MAC207 epitope in explants (arrow) and (L) inset in somatic embryos (double arrow). (**M**) Organ-like structure (arrow), calcofluor white staining. (**M’**) Same section as in M, lack of MAC207 signal, dotted line—outline of the organ-like structure. (**N**) MAC207 epitope observed in explant cells (double arrow) but not in the bulges (arrow); dotted line marks the border between explants and bulges. (**O**) Moderate intensity of fluorescence signal throughout the explant; inset—magnification of explant cells indicating the MAC207 signal in cytoplasm. (**P**) MAC207 signal detected in organ-like structures (arrow) and explant cells (double arrow). 6d—6th day of the culture; 14d—14th day of the culture. Scale bar: (**A**,**A’**)—50 µm; (**B**,**B’**)—40 µm; (**C**,**C’**)—100 µm; (**D**,**D’**,**H**,**I**)—200 µm; (**E**,**F**,**G**,**J**,**K**)—250 µm; (**L**), L inset, (**M**,**M’**,**P**)—100 µm; (**N**,**O**)—50 µm; O inset—20 µm.

**Figure 5 ijms-24-10356-f005:**
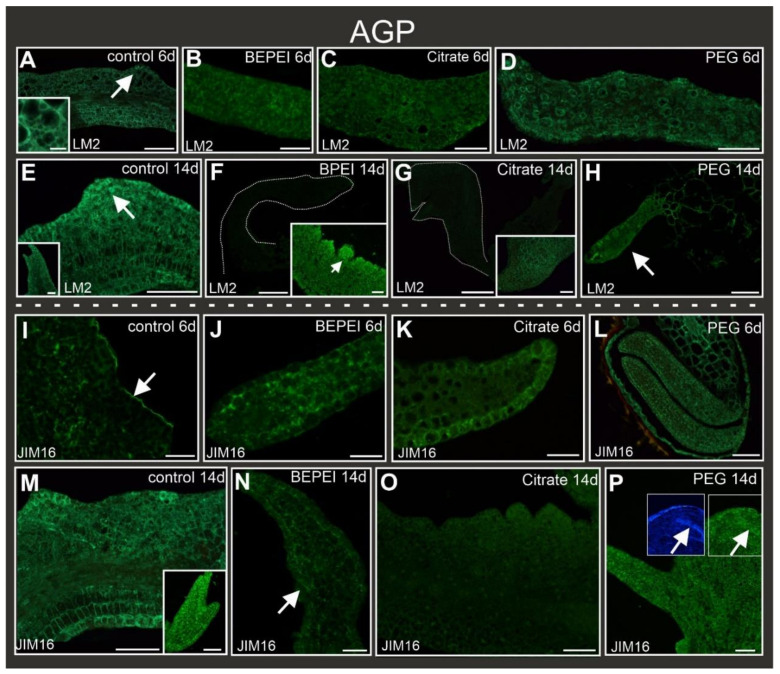
Localization of AGPs recognized by LM2 (**A**–**H**) and JIM16 (**I**–**P**) epitopes in control and Au NP-treated explants. A inset—higher magnification cells from protrusion area (marked by arrow). At the early stage of culture, the LM2 signal was present in cotyledons: (**A**–**D**). At the late stage of culture, the LM2 signal was present in (**E**)—explant cells and protrusions (arrow) and inset –somatic embryos. (**F**,**G**)—lack of LM2 signal in organ-like structures (outline denoted by dotted line), insets—LM2 detected in explant cells and bulges (arrow). (**H**)—LM2 fluorescence in organ-like structures (arrow) and explant cells. At the early stage of culture, the LM2 signal was present in cotyledons: (**I**–**L**); arrow—a periclinal wall of the epidermis. (**M**) JIM16 fluorescence was detected in the margin, protrusions, and explant cells; inset—intensive fluorescence signal in somatic embryos. (**N**) JIM16 signal in organ-like structure (arrow). (**O**) Lack of signal in organ-like structures in citrate Au NP-treated explants. (**P**) Image showing the JIM16 signal in organ-like structure (arrow indicates vascular bundle; P inset—vascular bundle after calcofluor staining). 6d—6th day of the culture; 14d—14th day of the culture. Scale bar: (**A**,**B**,**C**,**E**), E and F inset, (**I**–**P**), M inset—100 µm; A inset—20 µm; (**D**,**F**,**G**,**H**)—200 µm; (**K**),G inset—50 µm.

**Figure 6 ijms-24-10356-f006:**
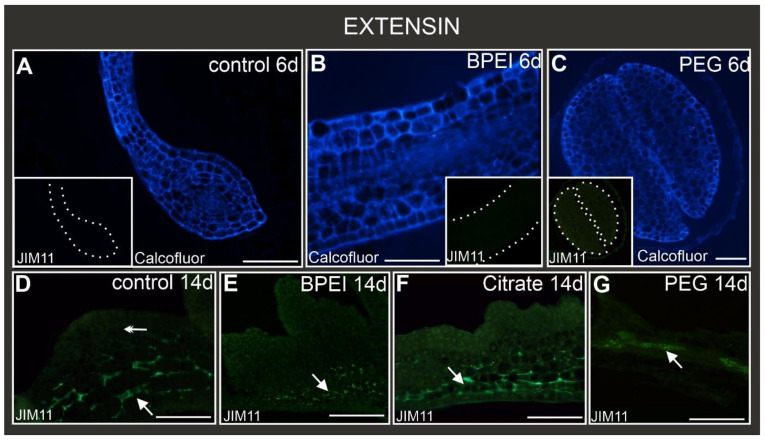
Localization of extensins recognized by JIM11 epitope in control and Au NP-treated explants. Calcofluor white staining of (**A**–**C**) at the early stage of culture. Insets A, B, C—corresponding sections observed under blue light showing no JIM11 signal. (**D**) JIM11 detected in explant cells (arrow) without visible fluorescence in margin cells and protrusions (double arrow). JIM11 was detected in explant cells in: (**E**–**F**) (arrows), but not in the surface cells or bulges. (**G**) JIM11 signal detected only in the vascular bundle (arrow). The dotted white line delimits the border of the cotyledons; 6d—6th day of the culture; 14d—14th day of the culture. Scale bar: (**A**–**G**)—100 µm.

**Table 1 ijms-24-10356-t001:** Percentages of control and Au NP-treated explants with morphological changes.

	N	Presence of Cotyledons Margin	Somatic Embryos	Bulge on Cotyledons	Bulge on Hypocotyl	Without Changes in Cell Fate
*35S:BBM*	146	90.4 a	84.2 a	2.1 a	0 a	11.0 a
*35S:BBM* + BPEI Au NPs	146	19.2 b	2.1 b	37.7 b	23.3 b	43.2 b
*35S:BBM* + Citrate Au NPs	144	21.5 b	0 b	25.0 b	20.8 b	50.0 b
*35S:BBM* + PEG Au NPs	144	9.0 c	0 b	34.0 b	16.7 b	46.5 b

The percentage values were obtained by counting the number of 14-day-old seedlings with developmental changes divided by the total number of evaluated seedlings. A total of 48–49 seedlings were evaluated in each of the 3 biological replications. The percentage values within a column followed by different letters are significantly different at *p* < 0.001 according to the χ^2^ test of contingency tables with count data.

**Table 2 ijms-24-10356-t002:** The distribution of the analyzed epitopes in the control explants and explants treated with different Au NPs.

		LM5	LM6	JIM13	MAC207	LM2	JIM16	JIM11	LM5	LM6	JIM13	MAC207	LM2	JIM16	JIM11	LM5	LM6	JIM13	MAC207	LM2	JIM16	JIM11	LM5	LM6	JIM13	MAC702	LM2	JIM16	JIM11
		EXPLANT CELLS							MARGIN CELLS							PROTRUSION							SOMATIC EMBRYOS						
**CONTROL**	EARLY	**+**	**+**	**−**	**−**	**+**	**+**	**−**	**+**	**+**	**+**	**−**	**+**	**−**	**−**	**+**	**+**	**−**	**−**	**+**	**−**	**−**	Not applicable						
	LATE	+	+	−	−	+	+	+	+	+	−	−	+	+	−	+	+	−	−	+	+	−	+	−	−	+	+	+	−
		**EXPLANT CELLS**							**SURFACE CELLS**							**BULGES**							**OGRAN-LIKE STRUCTURE**						
**BPEI**	EARLY	**+**	**+**	**−**	**−**	**+**	**+**	**−**	Not applicable							Not applicable							Not applicable						
	LATE	**−**	**+**	**−**	**−**	**+**	**+**	**+**	**+**	**−**	**−**	**−**	**+**	**+**	**−**	**+**	**−**	**−**	**+**	**+**	**+**	**−**	**−**	**+**	**−**	**−**	**−**	**+**	**−**
**CITRATE**	EARLY	**+**	**+**	**−**	**+**	**+**	**+**	**-**	Not applicable							Not applicable							Not applicable						
	LATE	**−**	**+**	**−**	**+**	**+**	**−**	**+**	**−**	**+**	**−**	**−**	**−**	**−**	**−**	**−**	**+**	**−**	**−**	**+**	**−**	**−**	**−**	**+**	**−**	**−**	**−**	**−**	**−**
**PEG**	EARLY	**−**	**+**	**−**	**−**	**+**	**+**	**−**	Not applicable							Not applicable							Not applicable						
	LATE	**−**	**−**	**−**	**+**	**+**	**−**	**−**	**+**	**−**	**−**	**−**	**−**	**−**	**−**	**+**	**−**	**−**	**−**	**−**	**+**	**−**	**+**	**+**	**−**	**+**	**+**	**+**	**−**

The “early” and “late” mean stages of culture. Plus means the presence of the analyzed epitope in the cells, and minus means the absence of the analyzed epitope in the cells. Because after six days of culture, no morphological changes in the explant were observed, the term “not applicable” was introduced.

**Table 3 ijms-24-10356-t003:** List of the monoclonal antibodies used in the current study, the epitopes they recognize, and references.

Antibody	Recognized Epitope	References
Pectins		
LM5	Linear tetrasaccharide in (1→4)-β-D-galactans (RG I side chain)	[84]
LM6	Linear pentasaccharide in (1→5)-α-L-arabinans (RG I side chain)	[85]
AGP		
JIM13	Arabinogalactan/arabinogalactan protein, carbohydrate epitope (β)GlcA1→3(α)GalA1→2Rha	[86]
LM2	Arabinogalactan/arabinogalactan protein, carbohydrate epitope containing β→linked GlcA	[87]
MAC207	Arabinogalactan protein, (β)GlcA1→3(α)GalA1→2Rha	[88]
JIM16	AGP glycan	[86]
Extensins		
JIM11	Extensin/HRGP glycoprotein	[89]

List of abbreviations: AGP—arabinogalactan proteins; Au NPs—gold nanoparticles; BPEI Au NPs—positively charged gold nanoparticles; citrate Au NPs—negatively charged gold nanoparticles; NPs—nanoparticles; PEG Au NPs—neutral gold nanoparticles; RG I—rhamnogalacturonan I; SE—somatic embryogenesis.

## Data Availability

The data presented in this study are available on request from the corresponding author. The data are not publicly available due to privacy.

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
