# Peer review of "The Impact of Gold Nanoparticles on Somatic Embryogenesis Using the Example of Arabidopsis thaliana"

_ijms, 2023, doi:10.3390/ijms241210356_

Round 1

Reviewer 1 Report

The manuscript entitled "The impact of gold nanoparticles on somatic embryogenesis on 2 the example of Arabidopsis" is interesting study. My concern is about statistical portion. And comprehensive details are require to update your methodology portion. I am recommending minor revision.

N/A

Author Response

Thank you very much for such a positive review. We have taken into account your comments. We appreciate the effort and detailed reading of the manuscript.

Reviewer

The manuscript entitled "The impact of gold nanoparticles on somatic embryogenesis on 2 the example of Arabidopsis" is interesting study. My concern is about statistical portion. And comprehensive details are require to update your methodology portion. I am recommending minor revision.

Answer

In the present version of MN, information about the statistical methods is described in the material and methods part and is visible in the review mode.

Moreover, the discussion has been changed. In the current version of the manuscript, the previous one has been deleted, and the new one has been inserted. All changes are visible in review mode.

Reviewer 2 Report

The manuscript describes the influence of gold nanoparticles on induction of somatic embryogenesis in transgenic line of Arabidopsis. This is an interesting study, but there are some suggestions for improving the manuscript.

L.32, 44. It is necessary to define the NP and SE at the first mention in the main text.

Please add information about the 35S:BBM transgenic line to the Introduction.

L.486. Please add a reference for the MS medium.

What is the difference between Fig. 1E and Fig. 1I? They demonstrated the cotyledons of control plants on the 14th day of culture.

Fig. 2C. Probably “control 14 d”, since “At the late stage of the culture, somatic embryos were present” (L.150-151).

Scheme 1 (should be Figure 3). "Outhgrowths" should be corrected to "outgrowths".

It is interesting to test the effect of NPs on the SE in a non-transgenic line.

Reference style does not meet the requirements of the journal.

Author Response

Thank you very much for your insightful, constructive review and comments. They were taken into account and certainly contributed to improving the quality of our publication. We appreciate the effort and detailed reading of the manuscript.

The manuscript describes the influence of gold nanoparticles on induction of somatic embryogenesis in transgenic line of Arabidopsis. This is an interesting study, but there are some suggestions for improving the manuscript.

Answer

We are very pleased that our research has been found interesting and worth publishing.

L.32, 44. It is necessary to define the NP and SE at the first mention in the main text.

Answer

It is changed and visible in the review mode.

Please add information about the 35S:BBM transgenic line to the Introduction.

Answer

It is added and visible in the review mode.

L.486. Please add a reference for the MS medium.

Answer

It is changed and visible in the review mode.

What is the difference between Fig. 1E and Fig. 1I? They demonstrated the cotyledons of control plants on the 14th day of culture.

Answer

The explanation is in the figure caption.

Fig. 2C. Probably “control 14 d”, since “At the late stage of the culture, somatic embryos were present” (L.150-151).

Answer

Thank you for this comment. It was our mistake, and now it is changed and visible in the review mode.

Scheme 1 (should be Figure 3). "Outhgrowths" should be corrected to "outgrowths".

Answer

Because one reviewer considered scheme 1 as a „nightmare”, we made changes and it is now Table 2. We hope that now this summary of results is clearer. Also, outgrowth has been changed to bulge because we felt it is more appropriate for the changes being described.

It is interesting to test the effect of NPs on the SE in a non-transgenic line.

Answer

Yes, we agree, and we plant to do such experiments, especially since we already analyzed following Arabidopsis (L.) Heynh Columbia-0 (Col-0) lines in in vitro culture: wild-type (WT), 35S:BBM, 35S:BBM WOX2:NLS-YFP, and 35S:BBM-GR Dr5v2tdTomato to study the somatic embryogenesis in context of somatic embryo development and involvement of symplasmic isolation in the initiation of cell fate changes.

Reference style does not meet the requirements of the journal.

Answer

In the present version of our manuscript it is done.

Moreover, the discussion has been changed. In the current version of the manuscript, the previous one has been deleted and the new one has been inserted. All changes are visible in review mode.

Reviewer 3 Report

The manuscript ijms-2425463 presents a study concerning the effects of gold nanoparticles on somatic embryogenesis in arabidopsis. The concept is clear and the basic conclusion, i.e. that gold nanoparticles inhibit somatic embryogenesis and promote organogenesis instead, is obvious, if we accept to believe the claims of the authors. Furthermore, cell reprogramming is combined with alterations in cell wall composition, related to its ability for expansion and/or cell division. However, documentation is not convincing and the whole manuscript requires much revision to be brought at an acceptable level. Several of my comments are annotated on the attached PDF. Here I summarize the most important ones:

1. In general, a non-specialist reader cannot discern embryos and organs in small figures like those in Fig. 1. The authors have to provide much more informative images. In addition, when mentioning "organogenesis" what organ do the "organ-like" structures represent?

2. Scheme 1 is just a nightmare! Please find a more eye-friendly way to summarize the results, this is confusing, almost unreadable.

3. Several immunofluorescent figures appear fuzzy, not exhibiting what they are supposed to. Please provide more clear and convincing documentation, also including -as a supplement?- the respective negative controls.

4. The intensity of immunofluorescence signals has to be quantified  so that differences may have validity.

5. The expected cell-wall-localized signal is sometimes cytoplasmic. How can you explain this?

6. There are problems with English style almost everywhere but Discussion has more: It appears that each paragraph has been written by somebody else, with no uniforrmity in style, nomenclature and symbols. In addition, the text has to be restructured to become more rigid and laconic, straight to the meaning.

7. I suggest that throughout the manuscript the plant is written as "Arabidopsis thaliana" and that the abbreviations NP and SE are aborted.

8. In the discussion, the comparison among other metal nanoparticles and gold ones does not lead anywhere. Please be more conclusive.

9. Why did you not also apply gold nanoparticles without any extra coating?

10. The calculation of mitotic index in each of the treatments would be also supportive for the conclusions.

The text requires English style corrections, as well as an effort to be less rambling. Tenses are not correct at several points etc. I have made some corrections myself, however the assistance of a native English writter with the appropriate scientific background would greatly improve the manuscript.

Author Response

Thank you very much for your insightful, constructive review and comments. They were taken into account and certainly contributed to improving the quality of our publication. We appreciate the effort and detailed reading of the manuscript.

Detailed responses to review along with the new version of MS are in the attachment.

Round 2

Reviewer 3 Report

The revised version of "ijms-2425463" is much better than the original submission: The new figures are convincing and the table that has replaced previous "scheme 1" is informative and easy to read. What has been also much better is the Discussion part. My only negative comments are about the NP and SE abbreviations that still exist throughout the text, as well as some random mistakes, (i.e. Arabidopsis (L.) Heynh, no full name of the species in Materials and methods) that have to be corrected. So, a final correction and "polishing" step is required before acceptance. In addition, a single negative control panel for the immunofluorescence, provided as supplement, would be useful.

I also have one question, concerning the response of the authors:

"The synthesis of pectins and AGP takes place in the AG." what is the "AG"???

As for the whole manuscript, a final "polishing" step of English style would make the text easier to be read.

Author Response

Thank you for re-reviewing and responding positively to our changes. The abbreviations used for NPs and SE have been explained in each of three sections: the abstract; the main text; the first figure or table, as is required by the journal. These are commonly used abbreviations in many publications, such as: (1) Horstman, A., Bemer, M., & Boutilier, K. (2017). A transcriptional view on somatic embryogenesis. Regeneration, 4(4), 201-216. (2) Wójcikowska, B., & Gaj, M. D. (2016). Somatic embryogenesis in Arabidopsis. Somatic embryogenesis: fundamental aspects and applications, 185-199. (3) Rose, R. J., & Nolan, K. E. (2006). Invited review: genetic regulation of somatic embryogenesis with particular reference to Arabidopsis thaliana and Medicago truncatula. In Vitro Cellular & Developmental Biology-Plant, 42, 473-481. (4) Tarrahi, R., Mahjouri, S., & Khataee, A. (2021). A review on in vivo and in vitro nanotoxicological studies in plants: A headlight for future targets. Ecotoxicology and Environmental Safety, 208, 111697. (5) Ndlovu, N., Mayaya, T., Muitire, C., & Munyengwa, N. (2020). Nanotechnology applications in crop production and food systems. International Journal of Plant Breeding, 7(1), 624-634. (6) Shrestha, S., Wang, B., & Dutta, P. (2020). Nanoparticle processing: Understanding and controlling aggregation. Advances in Colloid and Interface Science, 279, 102162 and literature herein. The use of the full name (SE and NPs) made it much more difficult to read and understand the main text, in our opinion. In particular, aborting the NPs abbreviation would make it necessary to enter the phrase "Citrate- Au nanoparticles, BPEI- Au nanoparticles or PEG-Au nanoparticles" for each antibody description in the results which makes this section very complicated. The text has been corrected and smoothed. In the materials and methods section, the full name of the species Arabidopsis thaliana has been added. In addition, we have added the figure, as mentioned earlier, showing the negative control as supplementary material. In connection with the question, AG means the Golgi Apparatus. So complete sentence would be: "The synthesis of pectins and AGP takes place in the Golgi Apparatus."